# Exploring the Practical Applications of Artificial Intelligence, Deep Learning, and Machine Learning in Maxillofacial Surgery: A Comprehensive Analysis of Published Works

**DOI:** 10.3390/bioengineering11070679

**Published:** 2024-07-03

**Authors:** Ladislav Czako, Barbora Sufliarsky, Kristian Simko, Marek Sovis, Ivana Vidova, Julia Farska, Michaela Lifková, Tomas Hamar, Branislav Galis

**Affiliations:** 1Department of Oral and Maxillofacial Surgery, Faculty of Medicine, Comenius University in Bratislava and University Hospital, Ruzinovska 6, 826 06 Bratislava, Slovakia; czako@ionline.sk (L.C.); kristian.simko@gmail.com (K.S.); mareksovis@gmail.com (M.S.); ivana.vidova94@gmail.com (I.V.); farskajulia@gmail.com (J.F.); brano.galis@gmail.com (B.G.); 2Department of Stomatology and Maxillofacial Surgery, Faculty of Medicine, Comenius University in Bratislava, St. Elisabeth Hospital Bratislava, Heydukova 10, 812 50 Bratislava, Slovakia; michaela.lifkova@fmed.uniba.sk; 3Institute of Medical Terminology and Foreign Languages, Faculty of Medicine, Comenius University in Bratislava, Moskovska 2, 811 08 Bratislava, Slovakia; tomas.hamar@fmed.uniba.sk

**Keywords:** artificial intelligence, deep learning, machine learning, maxillofacial surgery, evidence–based practice

## Abstract

Artificial intelligence (AI), deep learning (DL), and machine learning (ML) are computer, machine, and engineering systems that mimic human intelligence to devise procedures. These technologies also provide opportunities to advance diagnostics and planning in human medicine and dentistry. The purpose of this literature review was to ascertain the applicability and significance of AI and to highlight its uses in maxillofacial surgery. Our primary inclusion criterion was an original paper written in English focusing on the use of AI, DL, or ML in maxillofacial surgery. The sources were PubMed, Scopus, and Web of Science, and the queries were made on the 31 December 2023. The search strings used were “artificial intelligence maxillofacial surgery”, “machine learning maxillofacial surgery”, and “deep learning maxillofacial surgery”. Following the removal of duplicates, the remaining search results were screened by three independent operators to minimize the risk of bias. A total of 324 publications from 1992 to 2023 were finally selected. These were calculated according to the year of publication with a continuous increase (excluding 2012 and 2013) and R^2^ = 0.9295. Generally, in orthognathic dentistry and maxillofacial surgery, AI and ML have gained popularity over the past few decades. When we included the keywords “planning in maxillofacial surgery” and “planning in orthognathic surgery”, the number significantly increased to 7535 publications. The first publication appeared in 1965, with an increasing trend (excluding 2014–2018), with an R^2^ value of 0.8642. These technologies have been found to be useful in diagnosis and treatment planning in head and neck surgical oncology, cosmetic and aesthetic surgery, and oral pathology. In orthognathic surgery, they have been utilized for diagnosis, treatment planning, assessment of treatment needs, and cephalometric analyses, among other applications. This review confirms that the current use of AI and ML in maxillofacial surgery is focused mainly on evaluating digital diagnostic methods, especially radiology, treatment plans, and postoperative results. However, as these technologies become integrated into maxillofacial surgery and robotic surgery in the head and neck region, it is expected that they will be gradually utilized to plan and comprehensively evaluate the success of maxillofacial surgeries.

## 1. Introduction

Everyday surgical practice often requires the maxillofacial surgeon to make immediate and complex decisions in given surgical situations without jeopardizing the success of the procedure or the health of the patient. Such medical training demands critical, quick, and rational thinking; this, along with the logical progression of a surgical procedure, facilitates an algorithmic way of thinking. Every planned clinical and paraclinical procedure is analyzed, which subsequently allows for solving all unexpected and rare critical surgical situations [1]. The surgeon’s high workload often does not allow such a systematic approach, as it is very time-consuming and represents a significant mental burden on the surgeon themself. The decision-making process itself can be strongly influenced by the personal experiences, fatigue, mental state, or personality characteristics of the surgeon. Increased mental stress can subsequently lead to inaccuracies and even errors in all stages of patient treatment—diagnosis, therapeutic decisions, surgical procedures, and follow-up [2]. Because of advances and refinements in computing systems and algorithms, the exponential production and recording of health data, and the creation of large usable datasets, artificial intelligence (AI), deep learning (DL), and machine learning (ML) are rapidly evolving within the healthcare sector. All the mentioned technologies represent valuable support for medical reasoning, potentially limiting cognitive biases and, thus, also medical errors, especially in the diagnosis and planning of surgical treatment. Current and future AI methods and techniques promise to improve the surgeon’s practice in all phases of surgical patient management, including screening, diagnosis, therapeutic decision-making, surgical approach, the surgery itself, follow-up, etc. An increasing number of algorithms now surpass human mental abilities and play crucial roles in specific medical fields, for example, in detecting breast cancer, in mammograms, and in forensic medicine and anthropology (forensic pathology or forensic science). With the application of medical knowledge and expertise to legal matters in the criminal justice system, particularly in the investigation of crimes to determine the cause and manner of death, objective scientific evidence is provided to support legal proceedings [3,4,5].

## 2. Materials and Methods

### 2.1. The Framework, the Protocol, and Research Questions

This paper reviewed the literature registered in PubMed, Scopus, and Web of Science. The main inclusion criterion was that it was an original paper in English focused on “artificial intelligence in maxillofacial surgery”, “deep learning in maxillofacial surgery”, “machine learning in maxillofacial surgery”, “planning in maxillofacial surgery”, or “planning in orthognathic surgery”. Subsequently, basic questions were determined, which we aimed to answer. The reviewed questions were simple and clearly defined. This literature review answered the defined research questions by gathering and summarizing all empirical evidence that met the pre-specified eligibility criteria for the use of AI, DL, and ML in maxillofacial surgery and is intended for the practitioners in the maxillofacial community who prefer evidence-based practice (EBP).

Research questions:How many publications per year are focused on maxillofacial AI, DL, and ML utilization? Are they increasing? Identify the annual publication count (1992–2023).What is the focus of publications within the field of maxillofacial surgery, and what proportion of the focus has been devoted to specific specialized maxillofacial topics in the latest publications on AI, DL, ML, and planning in maxillofacial surgery from the first publication?If we include planning and computation in maxillofacial surgery as part of AI, DL, and ML, how many articles have been published, and when was the first article published?

This study involved three types of analyses:Quantitative assessment of maxillofacial AI, DL, and ML publications in the past three decades.Possibilities of using AI, DL, and ML within the field of maxillofacial surgery in contemporary literature from 1992 to the present.Number of publications that include maxillofacial planning and computed planning in online databases per year from 1965 onward.

### 2.2. Evaluating the Past Publications

To compare the prevalence of information on artificial intelligence and its use in maxillofacial surgery, we searched the medical databases PubMed, Scopus, and Web of Science. The search was focused on the publication title, abstract, and keywords. The search combination included “artificial intelligence maxillofacial surgery” or “deep learning maxillofacial surgery” or “machine learning maxillofacial surgery”.

Subsequently, we searched for terms that referred to planning in maxillofacial surgery. Our primary focus was on the occurrence of planning in the combination of title, abstract, and keywords: “planning maxillofacial surgery” and “planning orthognathic surgery”. Secondarily, we specified and searched for “computed planning maxillofacial surgery” and “computed planning orthognathic surgery”. The results were compared. Graphs were created and calculated in Excel (Microsoft corporation, Redmond, WA, USA. Microsoft Excel 2023). Bivariate Pearson’s analysis was performed for relations between years and the numbers of publications, calculated in IBM SPSS, version 26.0. (IBM Corp. 2019, Armonk, NY, USA: IBM Corp), in each category (AI, Dl, ML/planning, and computed planning).

## 3. Results

For AI, DL, and ML, 324 publications since 1992 were identified, with a continuous increase (excluding 2012 and 2013) and R^2^ = 0.9295 (Figure 1). The publications most often described their use in diagnosis, treatment planning, radiological assessment, surgical navigation, predictive analysis, virtual assistance, prosthetic analysis, and manufacturing technologies. However, a new chapter in human medicine and surgery was opened by the possibility of combining AI and robotic surgery in the future, using an autonomous treatment algorithm over which the surgeon would only perform a control role or supervision.

Pearson’s correlation analysis revealed a strong positive correlation between the year and the number of articles about AI, DL, and ML (r = 0.622, *n* = 21, *p* = 0.003).

AI was also used in orthognathic surgery planning and in computed planning, where the predominant use of AI, DL, and ML was found in publications on maxillofacial surgery. The total number of publications on planning in maxillofacial surgery and in orthognathic surgery was 7535, with the first publication in 1965 also showing an increasing trend (excluding 2014–2018) (R^2^ = 0.8642) (Figure 2).

Pearson’s correlation analysis also revealed a very strong positive correlation between the year and the number of computed planning articles (r = 0.879, *n* = 50, *p* < 0.001) and a strong positive correlation between the year and planning in total (r = 0.785, *n* = 59, *p* < 0.001).

The first publications with the keywords “computed planning maxillofacial” and “computed planning orthognathic” were found in 1972 with a total of 3122, also showing an increasing trend (R^2^ = 0.9362) (Figure 3).

Finally, we compared the number of publications with the keywords “planning orthognathic and planning maxillofacial” and “computed planning maxillofacial”. A significant difference was found in orthognathic planning, where, according to the current literature, AI, DL, and ML were used the most (Figure 4).

## 4. Discussion

The use of AI, DL, or ML in maxillofacial surgery is a relatively recent development. While AI, DL, and ML have been commonly used in other medical fields for several decades, their application in maxillofacial surgery or dentistry has only recently begun [6,7]. An account of the first use of AI, DL, or ML in maxillofacial applications, which was defined as computerized synthetic human cognitive function, was published in 1987 by Richter et al. Richter and their team applied a model to determine the optimal diagnostic approach for comprehensive chronic cholecystitis management strategies. A computer model was used to measure the frequency of appropriate surgery, inappropriate surgery, complications, death, and medical costs [8]. The first direct application of AI in maxillofacial surgery was subsequently described by Stoker et al. in 1992. During this time, the first stereolithographic model was constructed using AI with the application of computer graphics. The technology was employed using standardized cephalometric analyses in one case. The stereolithographic models constructed from digital image data (computed tomography (CT) and magnetic resonance imaging (MRI)) allowed the surgeon to view the external and internal anatomy prior to surgery [9]. The common use of AI in maxillofacial surgery is a computer-aided surgical simulation system for orthodontics and orthognathic surgery. This system allowed surgeons to create a virtual surgical treatment plan and simulate the outcome of the surgery preoperatively, which, in turn, helped improve the accuracy and outcome of the surgery. Since then, AI, DL, and ML have been used for various tasks in dentoalveolar and maxillofacial surgery, including surgical anatomy, diagnosis, treatment planning, and intraoperative navigation. AI has been used to develop algorithms for the automatic detection and segmentation of anatomical structures in medical images, such as preoperative CT and MRI scans, which can aid in the diagnosis and planning of surgical treatment of maxillofacial conditions [10]. Artificial intelligence is used to develop modern surgical navigation systems and procedures that can help surgeons in real time to refine and speed up surgical performance.

These computerized systems use not only preoperative images (CT and MRI) and intraoperative data but also 3D analysis of the soft and hard parts of the orofacial system to provide the surgeon with real-time feedback on the position and orientation of surgical instruments, as well as the location of critical anatomical structures. The aesthetic and functional predictability of the surgical performance can also be provided [11]. As AI technology continues to evolve, it is likely to play an increasingly important role in modern maxillofacial surgery in the future.

To fully exploit the potential of AI in maxillofacial surgery, it is important for maxillofacial surgeons to understand the technical elements of AI and its possible applications in the head and neck region. The aim of our review is to summarize the current use possibilities and the application of AI in diagnostics, surgical anatomy, preoperative planning, navigation, and operative procedures in maxillofacial surgery. Medical education and training promote the development of critical, rational, and logical thought processes, contributing to an algorithmic approach to reasoning. By carefully analyzing and considering every clinical and paraclinical aspect, healthcare professionals can effectively manage even the most unforeseen and rare situations [12]. AI is used in maxillofacial surgery in various ways to improve patient outcomes and surgical efficiencies, such as in diagnosis and treatment, surgical navigation, predictive analytics, virtual AI-powered peri-surgical assistance, prosthesis and epithesis design, manufacturing, visualization, and prediction of after-surgery results for the patient’s better understanding of the procedure.

### 4.1. Diagnosis and Treatment

The application of artificial intelligence (AI) and machine learning (ML) in maxillofacial surgery has the potential to revolutionize diagnosis and treatment planning. Advanced technologies can amplify accuracy, enhance patient outcomes, and streamline clinical workflows [13]. AI algorithms assist in the diagnostic process by analyzing large volumes of data, such as X-rays or other images and patient records, and by identifying patterns and relationships that may be difficult for human clinicians to detect [14]. Machine learning algorithms can likely be taught to identify irregularities in radiographic images, resulting in improved diagnostic accuracy and more precise treatment strategies in daily clinical practice [15]. In the article by Zhang et al., five uses of machine learning methods in radiological images were examined: image segmentation, computer-assisted detection and diagnosis, functional brain research and neurological disease identification, image classification and retrieval, and image registration. Machine learning methods are employed in computer-assisted systems to support radiologists in daily diagnostic tasks, research, and practice, and the application of these techniques in radiology continues to develop [15]. Within the world of maxillofacial surgery, AI might aid in identifying specific anatomical features, further improve preoperative preparation, and minimize the likelihood of surgical complications. In oral surgery and dentistry, AI models have shown significant promise in recognizing implant types, predicting the success of implants, and optimizing implant designs [16]. The unintentional, excessive use of “decision shortcuts” made possible by the inquisitive approach gives rise to cognitive biases, which represent a way of thinking that systematically and predictably leads to errors in judgment under particular circumstances [17]. Although AI and ML as diagnostic tools might already be part of daily practice, the responsibility to choose the correct information and make the final right decision should remain with the medical practitioner.

### 4.2. Surgical Navigation

Utilizing artificial intelligence for surgical navigation constitutes one potential direction for its implementation in surgical fields, including oral and maxillofacial surgery.

Novel surgical navigation systems are emerging, incorporating AI technology to varying extents. The principle of surgical navigation using a stereotaxic system was developed at the beginning of the 20th century and described in 1908 by Horsley and Clarke [18]. Surgical navigation systems have undergone development and improvements, and different versions based on various concepts have been created. The availability of patient-specific data and their use were realized after the introduction of CT and MRI in the 1980s [19]. Subsequently, surgical navigation systems have been developing, and clinicians in different fields, such as oncology, traumatology, and neurosurgery, have been performing surgical procedures with the aid of navigation. Specifically in maxillofacial surgery, the use of surgical navigation can be traced back to 1994, when it aided the removal of a skull base tumor. Since then, surgical navigation has become an essential tool for surgical procedures for trauma or oncological and reconstructive cases [20,21,22,23,24,25,26].

A theoretical overview exploring the history of surgical navigation and its use in the clinical setting was published by Novelli et al., affirming that surgical navigation effectively improves oral surgery [24].

Advances in computing speed, an exponential increase in data load, and routine data collection have expedited the rapid progress of AI in the clinical healthcare industry. One of the newer surgical navigation systems utilizes augmented reality and artificial intelligence (ARAI). The ARAI system comprises a display mechanism suspended over the surgical field projecting 3D medical images specific to the patient’s anatomy. The results of cadaveric research in spinal surgery indicated that the surgical navigation system precisely identified starting points, and the overlay of the virtual anatomy was accurately aligned with the actual anatomy. Systems using AI and augmented reality could increase the effectiveness of minimally invasive surgery, also increasing the speed of the procedure [25].

A different study suggests that using surface registration based on automated machine learning (ML) enhances the precision of the image-guided surgical navigation system. The system involves utilizing a neural network model to generate a new point cloud that corresponds to the facial data collected by the passive probe of an optical tracking system, which is extracted from facial information obtained using computerized tomography. Such an approach allows the acquisition of the accurate registration and alignment of the data, facilitating improved and faster diagnosis, treatment planning, and surgical outcomes in various medical fields, including maxillofacial surgery.

The cutting-edge surface registration concept utilizes a neural network model to generate a new point cloud that corresponds to facial data collected by the passive probe of an optical tracking system (OTS). This new point cloud is extracted from the facial information obtained using computerized tomography. Such an innovative approach allows for more accurate registration and alignment of the patient’s facial data, facilitating improved diagnosis, treatment planning, and surgical outcomes in various medical fields [26]. Maxillofacial surgery is undoubtedly a surgical field that will benefit from the use of AI, ML, and augmented reality in different surgical procedures, such as trauma surgery, oncological surgery, and reconstructive or orthognathic surgery.

### 4.3. Predictive Analytics

Machine learning and predictive analytics using AI learning algorithms to analyze patient data offer the possibility of predictive insights, recommendations, multiple points of view, and assistance in diagnostic, surgical, and therapeutical decision-making. Algorithms should be able to collect, combine, and analyze a patient’s historical data, general data, and real-time information and data to identify patterns and guide the surgeon toward effective actions. AI is becoming an omnipresent component in different fields, whether medical, technical, or educational. Detecting radiographic alterations is a notable application of AI and a useful tool in predicting results in dental implant cases with peri-implantitis complications [27]. Among the many other potential applications of radiographic studies analysis is the field of head and neck oncology. Early prediction, diagnosis, and prognostic prediction of malignant lesions can be implemented in the robotics field together with surgical tools such as guided resection, enhancing safety and reducing the possibility of human error [28].

The actual use of neural networks in maxillofacial surgery is discussed in a study by Nayans Jha et al. [29]. The study summarized the current literature on the application of AI technologies to diagnose various subtypes of temporomandibular joint (TMJ) disorders, also analyzing the quality of the studies and evaluating the diagnostic precision of current AI models.

Study findings suggest that AI algorithms designed for automated temporomandibular disorder (TMD) diagnosis can serve as a decision support tool in conjunction with medical diagnostic imagining techniques, input data types, and other features. Case–control studies indicated a high risk of bias in patient selection. Small datasets and a lack of external validation were found in most of the studies. Further research, larger datasets, and greater accuracy can ensure the efficacy of developed models [29].

The current clinical use of AI is described in an article by Alicia Dean et al. describing computer-assisted and navigated piezoelectric surgery (CANPS). CANPS is a surgical technique comprising a piezoelectric device and surgical navigation working synergistically. It integrates the benefits of piezo surgery and navigation, providing continuous monitoring of the piezoelectric device tip. The combination of real-time monitoring of the device with monitoring and navigation allows the surgeon to proceed without the necessity of direct observation [30].

Along with oncology, radiology, and TMD analysis, AI has great potential in aesthetic surgery. By helping the surgeon with a proper diagnosis, analysis of soft and hard tissues, and big data collection for surgical possibilities, AI might predict the surgical result, helping the surgeon with pre- and perioperative surgical decisions. For superior outcomes, AI combined with human surgical intuition may be the best tool [31,32,33,34,35,36].

AI and machine learning have the potential to process large data volumes to achieve precise results. Their speed, accuracy, and effective analysis have a big advantage over human efficiency. Hypothetically, even minor changes in radiography or any other test results can be identified, and AI might detect pathologies earlier than humans.

However, the application of AI and ML in predictive analytics raises numerous ethical issues.

In the field of aesthetical medical care, the use of AI to classify attractiveness and beauty can raise ethical issues. Potential discrimination based on gender and ethnicity could lead to the spread of racial division and reduce diversity in aesthetical cosmetic surgical procedures [34].

To improve diversity and enhance applicability, gathering datasets from a wide range of sources, including various ethnicities, genders, and ages, is required. Currently, the lack of representation of black patients and providers in blepharoplasty or rhinoplasty procedures impacts the accuracy of algorithms [35]. Measurements acquired from AI provide only the numerical expressions of opinions and subjective evaluations of publishing healthcare providers. AI’s definition is therefore subject to cultural and personal influences [36].

In the future, when AI’s predictive analysis can be fully implemented and trusted in everyday clinical practice, the data collection for the whole population, different ethnicities, and surgical procedures will be wide, complex, and organized. The predictive analysis of AI and ML can assist in decision-making and act as a useful tool.

### 4.4. Powered Virtual Assistance

Virtual assistance, which is common in daily life, comprises a voice-activated virtual assistant, such as Alexa, Google Assistant, Siri, and others. User-friendly, efficient AI virtual assistants in smartphones, laptops, cars, and smart home systems are already available and efficiently implemented.

The use of a virtual assistant based on AI was studied and discussed in an article by Jyoti Mago et al. [37]. The study evaluated the usefulness of four voice-based virtual assistants in oral and maxillofacial radiology report writing. The findings of this small study indicated that while AI assistants were helpful in providing responses to questions, there is still significant room for growth in terms of the topics and information delivered. Another potential use of AI assistance is in enhancing patient management and organizational workflows. During the COVID-19 pandemic, chatbots provided automatic triage for acute cases, support management, and referral assistance [38]. Implementation of a virtual assistant (VA) providing guidelines, instruction, and navigation is found in healthcare organizations such as WebMD [39], Mayo Clinic’s First Aid [40], and others. With the increase in telehealth services and their use, AI-based virtual assistants can extend the capability of the medical professional workforce by ensuring safety for both patients and medical staff. Conversational chatbots have not only the potential to be standard tools during a pandemic but could serve as reinforcements for routine daily clinical work [38].

### 4.5. Prosthetic Design and Manufacturing

Digital technologies are becoming increasingly standardized and integrated into routine daily treatment protocols [41,42]. In the prosthodontic field, clinical laboratories have already incorporated the daily digital technology practice by using computer-aided design/computer-aided manufacturing (CAD/CAM) [42,43]. A systematic review of articles about the use and performance of AI in prosthodontics was published by S. Bernauer et al. The review included a relatively low number of studies and an honest overview of the current state and latest developments. The review findings suggest that incorporating AI in prosthodontics is conceivable for clinicians. However, the implementation and practicality will depend on economic feasibility and demand [44]. In oncological and severe facial trauma surgical cases, even after local or microvascular flap reconstruction, patients need prosthetic rehabilitation to improve their psychological and social comfort. Maxillofacial prosthetic rehabilitation is a procedure that restores function and aesthetics. The associated aesthetic and psychological issues demand that high-quality prosthetic restoration be a part of the whole treatment [45]. Prior to CAD/CAM technology, skilled hand-carving of a wax cast was necessary for prosthetic facial reconstruction. Advances in computer development have made digitalization of the process and digital design possible [46,47]. Collaboration between medical professionals, skilled computer engineers, and an AI program should remain an essential part of the process. However, along with the benefits, there is also the potential liability of the diagnostic and creative role becoming too reliant on an AI system [48].

### 4.6. Orthognathic Surgery, Analysis and Prediction

Facial, dentofacial, and skeletal anomalies and irregularities have a negative impact on an individual’s well-being. Orofacial appearance is crucial to an individual’s social well-being and oral health-related life quality [49]. Facial appearance is affected by the skeletal framework. Facial anomalies can thus reflect the skeletal irregularities that might require treatment for dental occlusion. In many instances, it is often impossible to resolve the problem with orthodontic treatment alone, and orthognathic surgery is also necessary [50]. For successful orthognathic surgery, precise preoperative planning is essential. Currently, 3D planning is the golden standard, comprising a CT scan, 3D facial photography, intraoral scanning, and 3D analysis. Some clinics continue to use only 2D preoperative analysis and dental impressions for casts, followed by manual models for surgery preparation. Without a detailed 3D analysis of soft tissues and hard tissues or airways, this leads to large discrepancies and missed details [51]. Three-dimensional virtual surgical planning offers more detailed surgical options, resulting in a complex plan and improvement in the quality and effectiveness of surgery [50,52]. ML, as a subset of AI, is applied frequently to improve diagnostic computer support. The process involves integrating algorithms into machines, allowing them to learn from data, make predictions, and solve problems without the need for human input [53]. The use of ML and AI for analysis, assessment, surgical prediction, and proposal of therapeutic strategy might improve the treatment of skeletal anomalies and orthognathic surgery. Modifying personal appearance requires a conversation between the doctor and patient, and the doctor must listen to the patient’s expectations, using not only a purely scientific approach but also an artistic eye and empathy to create a new aesthetical and functional face. Using AI in virtual planning, creation, and approach can benefit the patient’s skeletal anomalies.

The article “Pierre Robin Sequence and 3D Printed Personalized Composite Appliances in Interdisciplinary Approach” presents an option for the treatment of a congenital condition called Pierre Robin Sequence.

This condition affects the development of the lower jaw, tongue, and airway and thus affects breathing, feeding, and speech. Standard treatment involves surgery and, in more severe cases, tracheostomy, the use of external devices, and mandibular distraction. The abovementioned treatments are associated with significant morbidity and might not always provide optimal results. The use of virtual analysis and creation, together with 3D printing technology that can customize personalized appliances, offers a cost-effective, time-effective treatment [54]. AI, ML, and the virtual world can improve craniomaxillofacial surgery for adult and pediatric patients if used correctly.

### 4.7. Maxillofacial Robotic Surgery

In the future, the combination of AI, ML, DL, and robotic surgery has the potential to bring about significant progress in both the didactic and technical execution of surgical procedures in the maxillofacial region. Integrating AI with robotics can enhance surgical precision, improve outcomes, and revolutionize the field of maxillofacial surgery [55].

Robotic surgery, also known as robot-assisted surgery, involves the use of robotic systems to assist surgeons during surgical procedures. These innovative systems typically consist of robotic arms equipped with surgical instruments and a control console from which the surgeon operates. The surgeon’s movements are translated into precise movements of the robotic arms, allowing for enhanced dexterity, stability, and maneuverability in head and neck surgery [56,57].

According to a study conducted by Yanice H. Yang et al., AI technology is not yet capable of surpassing human surgeons in terms of performance. However, the study also revealed that AI algorithms based on computer vision can generate results that align with the assessments of expert surgeons in terms of technical proficiency. This suggests the potential for AI to contribute to the enhancement of surgical skills by providing a standardized approach for evaluating surgical techniques [56].

The integration of AI, DL, ML, and robotics in maxillofacial surgery holds great promise and opens up new possibilities for advanced surgical techniques and decision-making processes. However, there remain challenges to overcome other than the skill set. These include ensuring data privacy and security, addressing ethical concerns, validating AI algorithms for surgical safety and efficacy, and providing comprehensive training for surgeons to effectively utilize AI and robotic systems [55].

In conclusion, the union of AI and robotic surgery promises to advance the field of maxillofacial surgery. By leveraging AI to assist in surgical planning, navigation, instrument guidance, decision support, and training, surgeons can enhance their technical capabilities and improve patient outcomes. Continued research and development in this area will pave the way for a future where AI-powered robotic surgery is a viable alternative in maxillofacial procedures.

The exploration of practical applications of AI, DL, and ML in maxillofacial surgery unveils promising opportunities for enhancing surgical practices [6,7]. However, along with these advancements come significant challenges that need to be effectively addressed to ensure the safe and ethical integration of AI technologies in this specialized field.

One of the critical challenges revolves around ethical considerations, particularly regarding patient privacy and data security. As these technologies rely on sensitive patient data, robust measures must be implemented to safeguard confidentiality and protect patient information [12]. Collaborative efforts between technology developers, healthcare professionals, and regulatory bodies are essential to establish guidelines and protocols that uphold ethical standards in the use of AI in maxillofacial surgery [58,59]. The current focus of AI-device approval predominantly centers on technical performance validation rather than directly assessing the effectiveness of AI in improving patient care and outcomes. Moreover, the complexity of evaluating the generalizability of AI algorithms poses a challenge during the device approval process. Once AI devices receive approval, the onus falls on healthcare professionals to ascertain the real-world impact and benefits of the approved AI algorithms in clinical practice [60].

As AI continues to advance in medicine, there is a growing need for stringent regulations regarding data storage, safety, and data liquidation to ensure patient privacy and data security. Establishing clear guidelines and protocols for managing AI-generated data will be essential in fostering trust and acceptance among patients and healthcare providers. Moreover, promoting collaboration and synergy between physicians and AI systems will be crucial in harnessing the full potential of AI technology in improving healthcare outcomes. The partnership between human healthcare providers and AI tools can lead to more accurate diagnoses, personalized treatment plans, and streamlined patient care processes. By developing and adhering to strict rules and promoting effective collaboration, the future of AI in medicine holds great promise for revolutionizing healthcare delivery and patient experience [59,60].

## 5. Conclusions

In conclusion, while AI, DL, and ML technologies offer immense potential for advancing diagnostics, treatment planning, surgical precision, and patient outcomes in maxillofacial surgery, their responsible use is paramount.

Joint efforts, stringent validation processes, and ongoing education initiatives are essential to harness the benefits of AI while upholding ethical standards and complementing the expertise of healthcare professionals. By addressing these challenges thoughtfully and proactively, the integration of AI in maxillofacial surgery can pave the way for significant progress and enhanced patient care in the field.

## Figures and Tables

**Figure 1 bioengineering-11-00679-f001:**
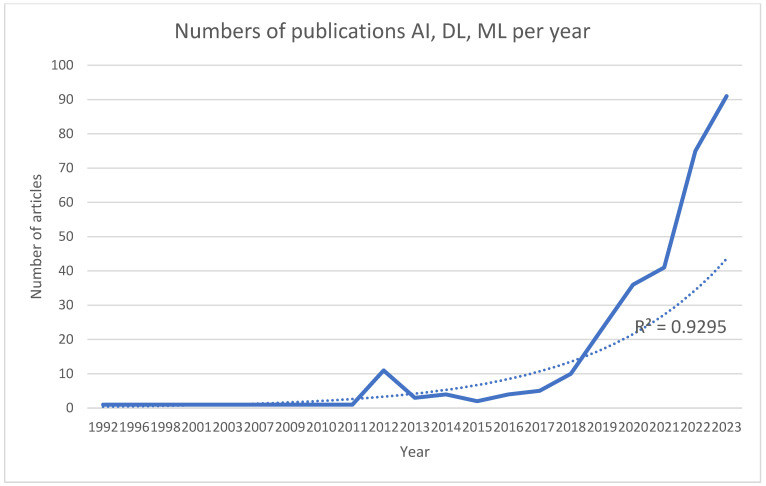
Numbers of publications per year with the phrases artificial intelligence, deep learning, and machine learning in maxillofacial surgery. The thick line represents raising number of publications and dotted line refers to R^2^.

**Figure 2 bioengineering-11-00679-f002:**
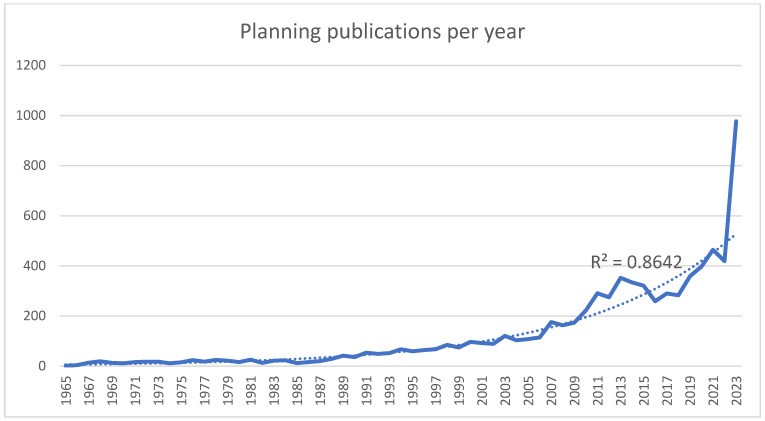
Numbers of overall publications on planning in maxillofacial surgery per year.

**Figure 3 bioengineering-11-00679-f003:**
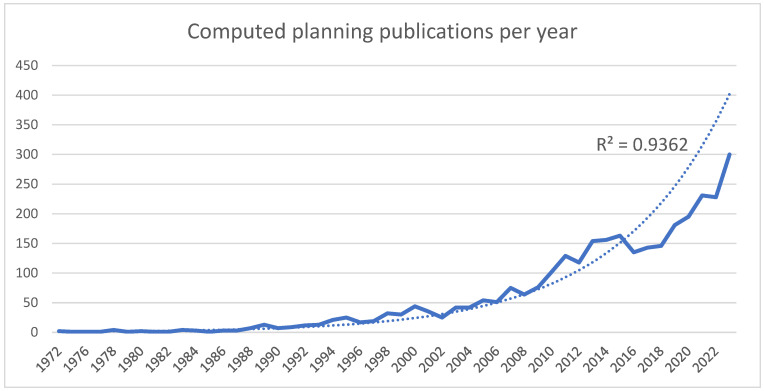
Numbers of computed planning publications in maxillofacial surgery per year.

**Figure 4 bioengineering-11-00679-f004:**
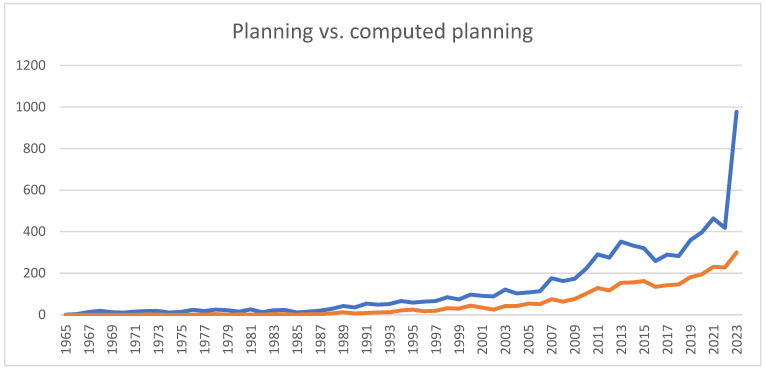
Comparison of the numbers of computed planning publications in maxillofacial surgery per year. The blue line represents planning publications, while the orange line represents the computed planning publications.

## Data Availability

All data and material are available on request from the authors.

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
