# Peer review of "Exploring the Practical Applications of Artificial Intelligence, Deep Learning, and Machine Learning in Maxillofacial Surgery: A Comprehensive Analysis of Published Works"

_bioengineering, 2024, doi:10.3390/bioengineering11070679_

Round 1

Reviewer 1 Report

Comments and Suggestions for Authors

The manuscript under review provides a broad literature review on the implementation of artificial intelligence in maxillofacial surgery. It is timely, covering a technologically advancing area with the potential to significantly impact surgical practices.

The document's length and repetitive nature could be improved. Statistical interpretations, particularly related to the trends in AI research within maxillofacial surgery, require further clarity. The literature review methodology would benefit from more detailed descriptions to assess potential biases. The paper needs a more structured approach in predicting future trends and research directions.

A critical appraisal of the literature is sometimes lacking, with some sections reading more as a list than an analytical review. The conclusions section should succinctly encapsulate key findings and propose clear directions for future research. Finally, attention to detail is needed to correct typographical and grammatical errors.

Condense the manuscript to eliminate redundancies and improve readability. Enhance the methodological clarity of the literature review and provide a more critical analysis of the findings. The conclusions should be concise, synthesizing critical insights and outlining future research pathways. Please address all language issues in the revised manuscript.

The subject of the manuscript is significant, and with the suggested modifications, it could be a notable contribution to maxillofacial surgery literature. The authors should make the necessary revisions and resubmit for re-evaluation.

Comments on the Quality of English Language

 Extensive editing of English language required

Author Response

Thank you for your insightful review of the manuscript. We have carefully addressed the feedback provided to enhance the article's quality. Our revisions include condensing the content to eliminate redundancies and improve readability. We have focused on clarifying statistical interpretations, especially regarding AI trends in maxillofacial surgery research. Additionally, we have refined the literature review methodology by providing more detailed descriptions to mitigate potential biases.

Furthermore, we have strengthened the critical analysis of the literature, ensuring it does not merely list information but offers an analytical perspective. The discussion has been added and conclusions now succinctly summarize key findings and suggest clear future research directions. We have also paid close attention to correcting typographical and grammatical errors throughout the manuscript.

We believe that with these enhancements, the manuscript now offers a more robust and informative discussion on the implementation of AI in maxillofacial surgery. We appreciate your feedback and look forward to your re-evaluation of the revised article.

Reviewer 2 Report

Comments and Suggestions for Authors

This is an interesting manuscript that comprehensively examines the historical backdrop and research advancements in the application of AI in oral and maxillofacial surgery. It provides readers with a valuable insight into the historical evolution and progress in this domain. However, there are still several issues which should be addressed.

1. The introduction section appears to be rather lengthy. It is recommended that most of its content be shifted to the "Discussion" section to maintain the structural integrity of the paper.

2. Notably, the manuscript lacks a dedicated "Discussion" section. It is advisable for the author to add this section into the manuscript.

3. The "Conclusion" section could be more concise. Ideally, it should comprise one or two succinct sentences that highlight the key insights for readers to take away.

4. In the newly added "Discussion" section, the author should delve into the primary challenges encountered in the application of AI in oral and maxillofacial surgery and offer a systematic analysis. This will be immensely valuable for readers to comprehend the current landscape.

5. Lastly, it is recommended to incorporate an "Outlook" section to clearly articulate the future research directions and potential avenues for exploration.

So, major revision should be recommended for this manuscript.

Author Response

Thank you for your detailed feedback on the manuscript.

We have implemented several significant changes based on your recommendations. While we have made extensive revisions, we believe that the introduction section's focus on revieweing the literature and use of AI, DL and ML in maxillofacial surgery is setting the tone for discussion to focus on ehtics, data safety, and collection.

We have revised the "Conclusion" section to provide succinctly highlight key insights for readers aligning with the recommendations.

Round 2

Reviewer 1 Report

Comments and Suggestions for Authors

It is great to see that you have taken our comments seriously.

Comments on the Quality of English Language

Thank you for improving it.

Author Response

Thank you very much for your insights and advices.

Reviewer 2 Report

Comments and Suggestions for Authors

The author's revised version is better than the original. However, some of the issues raised last time were not addressed on board. Includes:

1.   The introduction section appears to be rather lengthy.  It is recommended that most of its content be shifted to the "Discussion" section to maintain the structural integrity of the paper.

2.  It is recommended to incorporate an "Outlook" section to clearly articulate the future research directions and potential avenues for exploration.

Therefore, major revision should be recommended for this manuscript.

Author Response

Dear Reviewer,
Thank you for your insightful feedback on our article regarding AI in CMFs. We greatly appreciate the time and effort you have put into reviewing our work. In response to your suggestions, we have made several changes to the article based on your recommendations.
Firstly, we have shifted a significant portion of the content from the introduction to the discussion section, as per your recommendation. This adjustment has helped streamline the flow of the article and provided a more cohesive structure.
Secondly, we have emphasized the discussion of current and future challenges related to ethics and responsibility, as we agree that these are crucial aspects to consider in the context of AI in CMFs. By giving these challenges more prominence, we aim to contribute to a more comprehensive understanding of the implications of AI in this field.
Lastly, we acknowledge your observation that the outlook section, which typically uncovers future research directions, may not be as detailed as expected. We understand that our primary focus was to review existing publications on AI in CMFs, with discussions centered around its usage and challenges. However, we will take your feedback into consideration for future follow-up studies.
Once again, we are grateful for your valuable feedback, and we believe that the changes made in response to your recommendations have enhanced the quality and clarity of our article.
Thank you for helping us improve our work.

Round 3

Reviewer 2 Report

Comments and Suggestions for Authors

The revised version is much better than the original version. It could be accepted for now.